# Dietary Supplementation of a New Probiotic Compound Improves the Growth Performance and Health of Broilers by Altering the Composition of Cecal Microflora

**DOI:** 10.3390/biology11050633

**Published:** 2022-04-21

**Authors:** Kai Qiu, Xiaocui Wang, Haijun Zhang, Jing Wang, Guanghai Qi, Shugeng Wu

**Affiliations:** National Engineering Research Center of Biological Feed, Institute of Feed Research, Chinese Academy of Agricultural Sciences, Beijing 100081, China; qiukai@caas.cn (K.Q.); maque3001@163.com (X.W.); zhanghaijun@caas.cn (H.Z.); wangjing@caas.cn (J.W.); qiguanghai@caas.cn (G.Q.)

**Keywords:** *Lactobacillus*, *Yeast*, probiotics, intestinal health, cecal microflora, broilers

## Abstract

**Simple Summary:**

In most countries, antibiotic growth promoters are restricted or banned in the livestock industry, and probiotics have been widely explored to replace them. *Lactobacillus* LP184 and *Yeast* SC167 were selected as probiotic strains that could remain viable in feed and the gastrointestinal tract and were combined to form a compound to act as a substitute for antibiotics in broilers’ diets. This study aimed to investigate the effects of the compound probiotics as a potential alternative to antibiotics in broiler production. The feeding trial contained three dietary treatments and lasted for 42 days. The negative control group was fed the basal diet. The positive control group was fed the basal diet supplemented with commercial antibiotics. The probiotics group was fed the basal diet containing the compound probiotics. The results showed that the compound probiotics were a competent alternative for synthetic antibiotics to improve the production of broilers. The compound probiotics enhanced the immune and antioxidant capacities of broilers, which could not be achieved using antibiotics. The positive effects of the compound probiotics on the growth performance and health of broilers can likely be attributed to the improvement of intestinal morphology and cecal microbial diversity, effects which are distinct from those of antibiotics. These findings demonstrate the feasibility of replacing antibiotics with compound probiotics in broilers’ diets.

**Abstract:**

The current study aimed to investigate the effects of a new probiotic compound developed as a potential alternative to synthetic antibiotics for broilers. A total of 360 newly hatched Arbor Acres male chicks were randomly divided into three treatment groups. Each treatment consisted of six replicates with 20 birds in each replicate. The negative control group was fed the basal diet. The positive control group was fed the basal diet supplemented with a commercial antimicrobial, virginiamycin, at 30 mg/kg of basal feed. The compound probiotics group was fed a basal diet containing 4.5 × 10^6^ CFU of *Lactobacillus* LP184 and 2.4 × 10^6^ CFU of *Yeast* SC167 per gram of basal feed. The feeding trial lasted for 42 days. The results showed that the compound probiotics were a competent alternative to synthetic antibiotics for improving the growth performance and carcass traits of broilers. The compound probiotics enhanced the immune and antioxidant capacities of the broilers, while antibiotics lacked such merits. The positive effects of compound probiotics could be attributed to an improvement in the intestinal morphology and cecal microbial diversity of broilers, effects which are distinct from those of antibiotics. These findings revealed the differences between probiotics and antibiotics in terms of improving broilers’ performance and enriched the basic knowledge surrounding the intestinal microbial structure of broilers.

## 1. Introduction

The extensive use of antibiotics as antimicrobial agents over the past fifty years has resulted in the emergence of resistant bacteria and drug residues in foods [1,2,3]. The reduction or ban on the use of synthetic antibiotics as growth promoters in feed are global trends in the livestock industry, and the adoption of non-antibiotic approaches to replace synthetic antibiotics and maximize economic benefits has been widely explored and applied. Broilers do not have fully developed immune systems, especially during the starter phase, thus making the birds highly susceptible to bacterial infection. The development of a stable intestinal micropopulation in broilers may take more than two months [4]. From a public health perspective, probiotics and prebiotics could be applied in poultry feed safely without negative effects on the physiological status of the birds [2]. Probiotics can enhance the growth and health of poultry via numerous mechanisms [5,6,7], and have been characterized as a viable alternative to antibiotics in the healthful rearing of broilers [8]. Probiotic candidates, including *Lactobacillus*, *Bifidobacterium*, *Bacillus*, *Saccharomyces*, and *Faecalibacterium isolates*, have exhibited promising potential in increasing animal gut health and food safety [9].

Although synthetic antibiotics may act as therapeutic agents, various species of probiotics that can stimulate immune system are considered as better and safer alternatives. The potency of probiotics over antibiotics in the treatment of necrotic enteritis and the improvement in the performance and immunological potentials of broiler birds have been explored [10,11]. *L. plantarum* S27 isolated from chicken feces was shown improve animal feed intake and weight [12]. In addition, several strains of *Lactobacillus*, including *L. acidophilus*, *L. crispatus*, *L. fermentum*, *L. gallinarum*, *L. johnsonii*, *L. plantarum*, and *L. salivarius,* have been explored and shown to be an effective alternative to antibiotics for maintaining the intestinal health and immune capacity of broilers [13,14,15,16,17]. Live *Yeast* and its derivatives as effective and harmless feed additives have been shown to significantly increase the growth performance of broilers by adjusting their cecal microbial structure, reducing the prevalence of *Salmonella* and *Escherichia coli*, and ameliorating the deleterious effects of coccidiosis [18,19,20,21]. In addition, some strains of *Yeast* have been shown to be tolerant to the adverse effects associated with low doses of aflatoxin B1 and ochratoxin A in broilers’ diets [22,23,24]. Above all, both *Lactobacillus* and *Yeast* are suitable alternatives to antibiotic growth promoters in the diets of broilers.

Since multiple strains of probiotics can induce a “synergistic effect” that enhances the survival of probiotic bacteria and maintains the balance and stability of the host gastrointestinal microbiota [8], the combination of *Lactobacillus* and *Yeast* was expected to achieve a good therapeutic result as an adjuvant or alternative therapy for enhancing antibacterial effects in broiler production. The antibiotic alternative used in the current study was a combination of *Lactobacillus* LP184 and *Yeast* SC167, both of which were recently selected as two exclusive probiotic strains with the capacity to remain viable in feed and tolerate the adverse conditions of the upper gastrointestinal tract. However, little is known about the potential of these new compound probiotics for use as commercial antimicrobial substitutes in the diet of broilers. Therefore, the current study was designed to evaluate the effects of the compound probiotic on the growth performance, carcass traits, antioxidant and immune capacity, jejunum morphology, and cecal microbial composition of broilers.

## 2. Materials and Methods

### 2.1. Animals and Experimental Diets

Three hundred and sixty newly hatched 1-day-old Arbor Acres (AA) male chicks with similar body weight (BW) vaccinated against Newcastle disease and infectious bronchitis (Shanghai Haili Biotechnology Co., Ltd., Shanghai, China) were randomly allocated into 3 dietary treatments. Each treatment consisted of 6 replicates with 20 birds in each replicate. The chicks were raised in 3-tier battery cages, with 10 birds in each cage (100 cm long × 80 cm wide × 40 cm high). A 23L:1D light regime was maintained each day. All the birds were raised in a room with environmentally controlled conditions in the Nankou experimental base of the Chinese Academy of Agricultural Sciences (Beijing, China). The room temperature was 33 °C for the first three days after hatching and then reduced by 3 °C/week until 24 °C. The birds were free to access to feed (cold pellet form) and water and were managed according to the recommendations of the AA Broiler Management Guide (Aviagen, Huntsville, AL, USA, 2009).

The two basal diets based on corn and soybean meal were formulated in accordance with the nutrient recommendations of the National Research Council (1994) and Chinese Feeding Standard of Chicken (Ministry of Agriculture of China, Beijing, China, 2004); the ingredient composition and nutrition levels are shown in Table 1. The feeding trial lasted for 42 days. The negative control group (NC) was fed the basal diets. The positive control group (PC) was fed the basal diets supplemented with a commercial antimicrobial, virginiamycin, at 30 mg/kg of complete feed. The compound probiotic group (LY) was fed the basal diets with the addition of 4.5 × 10^6^ CFU of *L**actobacillus* LP184 and 2.4 × 10^6^ CFU of *Yeast* SC167 per gram of complete feed. *Lactobacillus* LP184 and *Yeast* SC167 were recently explored by our research group and were shown to be two potential probiotic strains that remained viable in feed and the hindgut of broilers. Counts of *Lactobacillus* and *Yeast* in the complete feed were confirmed by plate counting.

### 2.2. Data and Sample Collection

Feed intake from day 1 to 21 and day 22 to 42 was recorded. The total BW of broilers per replicate was measured on days 1, 21, and 42. The average daily feed intake (ADFI, feed intake:days, g:day), average daily gain (ADG, BW gain:days, g:day), and feed conversion ratio (FCR, feed:gain, g:g) were calculated in replicates accordingly. One bird that had a BW close to the average BW of the replicate was selected from each replicate at days 21 and 42 for blood sampling and carcass measurement after a 12 h fast. Blood was collected from the wing veins and subsequently centrifuged at 3000× *g* and 4 °C for 10 min. Plasma samples were stored at −20 °C until the analysis of biochemical indexes. After blood sampling, the birds were offered the experimental diets for 6 h; then, they were stunned at 40 V and 400 Hz for 5 s using an electrical stunner, immediately exsanguinated, and de-feathered to determine carcass weight. After the head, paws, abdominal fat, and giblets were removed, the eviscerated weight of the bird was obtained. The breast muscles (pectoralis major and minor), leg muscles (thigh and drumstick muscles), and abdominal fat (abdominal fat and the gizzard leaf fat surrounding the cloaca) were weighed. About 2 cm of jejunum (medial portion posterior to the bile ducts and anterior to Meckel’s diverticulum) was cut off gently and fixed in 10% formalin for histomorphology analysis. The eviscerated yield ratio was calculated by dividing the eviscerated weight into the carcass weight. The weight indexes of the breast muscle, leg muscle, and abdominal fat were calculated by dividing their weight into the live body weight. The immune organs, spleen, and bursa of Fabricius were also weighed and their weight indexes were calculated as their weight/live body weight. The digesta samples in the cecum were aseptically collected in two sterile containers; one was frozen by immersion in liquid nitrogen and stored at −80 °C for DNA extraction and sequencing, and the other was transported to the laboratory on ice for culturing *E. coli* and *Lactobacilli*.

### 2.3. Jejunum Histomorphology

Fixed jejunum samples were embedded in paraffin to make histological sections that were stained with hematoxylin and eosin. Three sections cut vertically from villus enterocytes to the muscularis mucosa were selected for one sample. The vertical distance from the villus tip to villus–crypt junction level was taken as the intestinal villus height (VH), and the vertical distance from the villus-crypt junction to the lower limit of the crypt was taken as the crypt depth (CD). Ten loci per section were measured for VH and CD. The ratio of VH to CD was calculated as V/C. A microscope coupled with a Microcomp integrated digital imaging analysis system (Nikon Eclipse 80i, Nikon Co., Tokyo, Japan) was used for the measurements.

### 2.4. Chemical Analysis

As described in our previous report [25], the concentrations of total protein (TP), albumin (ALB), uric acid (UA), and creatinine (CRE), as well as the activities of alkaline phosphatase (ALP), alanine aminotransferase (ALT), and aspartate aminotransferase (AST) in plasma were detected using an automatic biochemical analyzer (model 7020, Hitachi, Tokyo, Japan) with the A045-3-2, A028-1-1, C012-2-1, C011-2-1, C009-2-1, and C010-2-1 kits, respectively. The concentrations of IgG, IgM, and IgA in serum were determined by double-antibody sandwich ELISA using the H106, H109, and H108 kits, respectively. The enzymatic activities of total superoxide dismutase (T-SOD) and glutathione peroxidase (GSH-PX), total anti-oxidative capacity (T-AOC), and concentration of malondialdehyde (MDA) in serum were determined using the A001-3-2, A005-1-2, A015-1-2, and A003-1-2 kits, respectively. Each sample was analyzed in triplicate strictly according to the manufacturer’s instructions. All of the kits were offered by the Nanjing Jiancheng Bioengineering Institute (Nanjing, China).

### 2.5. Counts of Viable E. coli and Lactobacilli

The existence of *E. coli* and *Lactobacilli* in cecal digesta was confirmed with counts of viable bacteria. Cecal samples were diluted 10-fold using 1% sterile buffered peptone broth solution and mixed well with a vortex mixer. Then, 0.02% peptone solution was plated onto MacConkey agar plates (Difco Laboratories, Detroit, MI, USA) and *Lactobacilli* medium agar plates (Medium 638; DSMZ, Braunschweig, Germany) to isolate and culture *E. coli* and *Lactobacillus*, respectively. The MacConkey agar plates were incubated at 37 °C for 1 day; thereafter, the plates were taken out of the incubator and counted immediately. The *Lactobacilli* medium agar plates were incubated at 39 °C for 2 days under anaerobic conditions and were taken out and counted immediately. Finally, microflora concentrations were expressed as log10 colony-forming units per gram of intestinal contents.

### 2.6. Illumina MiSeq Sequencing of Cecal Microflora

The analysis of cecal microbiota was performed using high-throughput sequencing with four main steps: DNA extraction, PCR amplification of 16S rDNA, sequencing, and data analysis. The genomic DNA of the cecal microbiota was extracted using a Power Soil DNA Isolation Kit (Mo Bio, Carlsbad, CA, USA) according to the manufacturer’s protocol. The V3 hypervariable region of 16S rDNA in microbial genomic DNA was amplified via PCR reactions with the following primers: 338-CCTACGGGAGGCAGCAG-35 (forward) and 502-ATTACCGCGGCTGCTGG-518 (reverse). The PCR reaction conditions were as follows: 5 min at 94 °C for initial denaturation, followed by 25 cycles of 30 s at 94 °C, 30 s at 48 °C, and 30 s at 72 °C, with a final extension step of 10 min at 72 °C. The PCR products were isolated from 2% agarose gels after electrophoretic separation and purified using a MinElute Gel Extraction Kit (Qiagen, Venlo, Limburg, The Netherlands). Equal amounts of purified PCR products were mixed to generate the final sequencing library after the ends were repaired and poly(A) was added. The sequencing of the library was conducted using Illumina MiSeq at the Beijing Centre for Physical and Chemical Analysis (Beijing, China). After sequencing, all barcodes were deleted and the quality of the reads was assessed. In order to manage random sequencing errors, sequences shorter than 100 bp with mismatches to the PCR primers or with more than 1 undetermined nucleotide and an average Phred quality of ≤25 were removed. The barcodes and primers were trimmed from the assembled sequences, and the trimmed sequences were uploaded to QIIME (v 1.9.1, http://qiime.org, accessed on 17 August 2021) to generate the water abundance table of each taxonomy and calculate the beta diversity distance. A cluster analysis of operational taxonomic units (OTUs) was conducted using the Uparse software (v 7.0.1090). Sequence classification annotation and alpha microbial diversity analysis were performed using RDP Classifier (v 2.11) and Mothur (v 1.30.2), respectively.

### 2.7. Statistical Analysis

The data analysis was conducted using SPSS Version 19.0 (SPSS, Chicago, IL, USA). After checking whether the data followed a normal distribution using the Shapiro–Wilk test, all of the data (*n* = 6) were submitted to one-way ANOVA, and means were compared using Tukey’s post hoc test. *p* ≤ 0.05 was considered as a statistically significant difference.

## 3. Results

### 3.1. Performance and Carcass Traits

The growth performance of the broilers is listed in Table 2. At beginning of the trial, the BWs of broiler chicks were similar (*p* > 0.1) between the three experimental groups. On day 21, the BW of broilers in the LY group was significantly higher (*p* < 0.05) than that in the NC; both were not significantly different from those in the PC group (*p* > 0.1). On day 42, a significant difference (*p* < 0.05) in BW between the three groups was observed, with an obvious increase from the NC to PC to LY groups. From day 1 to 21, the ADFI was similar between the three groups. Compared with the NC and PC, the ADG of broilers in the LY group was significantly increased (*p* < 0.05) and the FCR was significantly decreased (*p* < 0.05). From day 22 to 42, the ADFI of broilers in the LY group was more (*p* < 0.05) than that of the NC; however, it was not different (*p* > 0.1) from those in the PC group. The ADG of broilers in the LY group was increased (*p* < 0.05) compared to that of the NC and PC, and that of the PC was higher (*p* < 0.05) than in the NC. Relative to the NC and PC, the FCR of broilers in the LY group was decreased (*p* < 0.05), and no differences (*p* > 0.1) were observed between the NC and PC groups. During the whole period of the trial, the ADFI of broilers in the LY and PC groups showed a tendency (*p* = 0.098) to increase relative to the NC. The ADG of broilers in the LY group was more (*p* < 0.05) than in the NC and PC, and that in the PC was higher (*p* < 0.05) than in the NC. The FCR of broilers was reduced (*p* < 0.05) in the LY group compared to the NC and PC.

As shown in Table 3, the eviscerated yield of broilers in the LY group showed a tendency (*p* = 0.074) to increase compared to the in NC and PC. The percentage of breast muscle of broilers in the LY and PC groups was increased (*p* < 0.05) compared to the NC. The relative weight of the leg muscle was higher (*p* < 0.05) in the LY group than in the NC and PC. The ratio of abdominal fat was significantly reduced (*p* = 0.05) in the LY group relative to the NC.

### 3.2. Serum Biochemical Indexes and Antioxidant and Immune Capacity

Biochemical indexes, including ALB, ALT, AST, CRE, TP, and UA, were analyzed in the serum on days 21 and 42 (Table 4). The UA concentration in the serum of broilers on day 21 showed a tendency (*p* = 0.085) to decrease in the LY relative to the NC and PC groups. On day 42, the concentration of TP in the serum of broilers was higher (*p* < 0.05) in the LY group than in the NC and PC. No differences (*p* > 0.1) with regard to the levels of ALB, ALT, AST, and CRE on days 21 and 42 existed among the three groups.

The antioxidant indexes in serum are shown in Table 5. On day 21, the activities of GSH-Px, T-AOC, and T-SOD in the serum of broilers were higher (*p* < 0.05) in the LY group than those in the NC and PC groups, and T-SOD was higher (*p* < 0.05) in the PC than in the NC. The concentration of MDA in the serum of broilers in the LY and PC groups was decreased (*p* < 0.05) relative to the NC. On day 42, the activities of GSH-Px and T-SOD in the serum of broilers in the LY group were higher (*p* < 0.05) than those in the NC and PC. The levels of MDA in serum were lower (*p* < 0.05) in the LY group compared with the NC; however, they were not different (*p* > 0.1) from the PC. There were no differences in the serum concentration of T-AOC (*p* > 0.1) among the three groups.

As shown in Table 6, the concentrations of IgA, IgG, and IgM in the serum of broilers in the LY group were higher (*p* < 0.05) than those in the NC and PC on both day 21 and 42. The immune organ indexes of broilers are presented in Table 7. On day 21, the weight ratio of the bursa showed a tendency (*p* = 0.064) to increase in the LY group relative to the NC and PC. The weight indexes of the thymus and spleen were not different (*p* > 0.1) among three groups. On day 42, the weight indexes of the spleen and bursa of broilers in the LY group were significantly (*p* < 0.05) increased compared to the NC and PC, while no differences (*p* > 0.1) were observed in the thymus among the three groups.

### 3.3. Jejunum Morphology and Cecum Microbiota

As shown in Table 8, on day 21, broilers in the LY group showed a higher (*p* < 0.05) VH and VCR and lower (*p* < 0.05) CD in the jejunum than in the NC and PC. On day 42, the VH and VCR in the jejunum of broilers in the LY group were greater (*p* < 0.05) than in the NC and PC. There were no differences (*p* > 0.1) in the CD of the jejunum among the three groups.

The concentrations of *Escherichia coli* and *Lactobacillus* in cecal digesta were determined on days 21 and 42 via culture under the specified conditions and a colony count (Table 9). On day 21, the number of *Escherichia coli* and *Lactobacillus* and their ratio in the cecum of broilers were not influenced (*p* > 0.1) by dietary treatments. On day 42, The amount of *Lactobacillus* in the cecum of broilers in the LY group was greater (*p* < 0.05) than in the NC and PC. The number of *Escherichia coli* and its ratio to *Lactobacillus* in the cecum of broilers in the LY group was the least (*p* < 0.05) among the three groups and was less in the PC (*p* < 0.05) than in the NC.

The cecal microbial diversity of the broilers is presented in Figure 1. The Shannon, Simpson, and Chao1 indexes of OUT levels in the cecum microbiota were similar among the three groups, while the ACE index of OUT levels linearly increased from the NC to PC to LY groups. The PC1, PC2, and PC3 of the results of a principal coordinate analysis (PCoA) based on the weighted UniFrac distance calculated from the OUT abundance matrix were 34.39%, 15.91%, and 10.48%, respectively (Figure 2a). The results of partial least squares discriminant analysis (PLS-DA) showed that points from the same group were marked with ellipses and the three ellipses were separated without overlap between them, which indicated that the distinctiveness of three groups was significant. When samples in the same group are close to each other, this indicates fewer differences within the groups. The samples belonging to the NC or LY groups were closer to each other than those of the PC (Figure 2b).

The most differentially abundant taxa enriched in the cecal microbiota of broilers identified by linear discriminant analysis effect size (LEfSE) are shown in Appendix A. The top ten taxa can be listed in order as follows: p_Firmicutes, c_Clostridia, o_Clostridiales, f_Veillonellaceae, f_Ruminococcaceae, p_Bacteroidetes, c_Bacteroidia, o_Bacteroidales, f_Bacteroidaceae, g_Bacteroides, p_Verrucomicrobia, c_Verruco-5, c_Verrucomicrobiae, o_Verrucomicrobiales, f_Verrucomicrobiaceae, g_Akkermansia, p_Proteobacteria, c_Deltaproteobacteria, o_Desulfovibrionales, and f_Desulfovibrionaceae. To assess the differences in cecal microbiota among the three groups, taxonomic compositions were analyzed at the phylum, class, order, family, and genus levels (Appendix A). As shown in Figure 3a, at the phylum level, the relative abundance of Bacteroidetes in the cecal microbiota of broilers in the LY group was increased (*p* < 0.05), while that of Cyanobacteria was decreased (*p* < 0.05), compared to the PC. The cecal microbiota of broilers in the PC group showed an increased (*p* < 0.05) relative abundance of Euryarchaeota relative to the NC. At the genus level, the relative abundances of *Alistipes*, *Helicobacter*, and *Lactococcus* in the cecal microbiota of broilers in the LY group were increased (*p* < 0.05), while that of *Campylobacter* was decreased (*p* < 0.05), compared to the NC. The cecal microbiota of broilers in the LY group showed lower (*p* < 0.05) relative abundances of *Burkholderia*, *Coprobacillus*, *Eggerthella*, and *Sediminibacterium*, and a higher abundance of (*p* < 0.05) *Prevotella*, than those in the PC. The relative abundances of *Alistipes*, *Lactococcus*, and *Methanocorpusculum* in the cecal microbiota of broilers were lower (*p* < 0.05) and the abundance of *Eggerthella* was higher (*p* < 0.05) in the PC than in the NC.

## 4. Discussion

In recent decades, about seventy percent of therapeutic measures adopted to improve gut health and production in livestock sector have been reliant on the use of synthetic antibiotics. The excessive use of antibiotic growth promoters in the livestock industry has raised several issues of concern about antimicrobial resistance and food safety [26]. In accordance with the ban or restriction of antibiotic use in animal feed, antibiotic-free diets for animal production have been successfully adopted by the European Union, United States, South Korea, and China. Probiotics and prebiotics have been widely explored and incorporated into chicken feed as natural alternatives to antibiotics in the maintenance of poultry health [2]. Multiple strains of probiotics have been shown to enhance the survival of probiotic bacteria and maintain the balance of the intestinal microbiota [8]. Both *Lactobacillus* LP184 and *Yeast* SC167, used in the current study, were recently selected as two potential probiotic strains and were prepared to act as a substitute to antibiotics in broilers’ diets with the capacity to remain viable in feed and in the gastrointestinal tract. Therefore, it was hypothesized that the supplementation of the developed compound containing *Lactobacillus* and *Yeast* would be effective in enhancing the growth performance and health of broilers.

In one study, the dietary supplementation of probiotics was shown to be very effective in improving the performance of broilers [11]; *Lactobacillus* and *Yeast*. *L. plantarum* S27 from chicken feces were also shown to improve feed intake and weight gain, and thus could act as a potential alternative to antibiotics [12]. In another study, *L. plantarum* 16 enhanced the growth performance and intestinal health of broiler birds during the starter phase [27]. *Lactobacillus* and inulin offered to broilers increased nutrient utilization and growth performance [28]. Dietary supplementation with *L. plantarum* 15-1 and fructooligosaccharides was shown to increase broilers’ growth by enhancing intestinal health [29]. *L. johnsonii* 3-1 and *L. crispatus* 7-4 were shown to reduce abdominal fat deposition by regulating liver lipid metabolism and promote growth performance and ileum development in broilers [16]. The dietary addition of *L. plantarum* P8 improved the growth performance and intestinal health of broilers infected with coccidiosis through the regulation of gut microbiota [15]. *Yeast* cultures with improved nutritional properties as effective and harmless feed additives have been shown to significantly improve the growth performance and feed utilization efficiency of broilers by balancing the cecal microbial community [30,31,32]. A mixed *Yeast* culture containing *Saccharomyces cerevisiae* YJM1592 and *Kluyveromyces maxianus* TB7258 in diets promoted increased weight gain and decreased FCR in broilers [18]. The dietary supplementation of *Yeast* in broilers challenged by *Salmonella enteritidis* was shown to improve growth performance and reduce *Salmonella* infection [33]. *Yeast* cell walls can partially recover the growth performance and intestinal health of broilers with concurrent challenges of aflatoxin B1 and necrotic enteritis [24]. The dietary supplementation of *Yeast* and its derivatives can reduce the negative effects of *salmonella* lipopolysaccharide on broiler chicks, thus increasing the performance and meat yield [20]. In the current study, the stimulating effects of the compound probiotic containing *Lactobacillus* and *Yeast* on the growth performance of broilers were confirmed again; the treatment also improved weight gain and FCR compared to the commercial antimicrobials. In addition, the percentages of eviscerated yield, breast muscle, and leg muscle were increased and that of abdominal fat was decreased in broilers that were fed the compound probiotics. This is consistent with previous reports showing that *Yeast* and its derivatives can increase meat yield and reduce abdominal fat deposition in broilers [16,20]. Therefore, we deduced that the developed probiotic compound could act as a growth promoter and carcass yield enhancer; hence, it could be considered as a potent alternative to synthetic antibiotics in broiler production.

Probiotics have been shown to be safer than antibiotics in stimulating the immune system against the colonization of *Salmonella* [10]. *Lactobacillus* and inulin fed to broilers were reported to strengthen the immune system [28]. The dietary supplementation of *L. plantarum* 15-1 was shown to enhance the immune response and mitigate intestinal damage caused by *E. coli* O78 [29]. *L. johnsonii* BS15 improved blood parameters related to immunity, enhanced intestinal immunity, and lessened hepatic inflammation effects in broilers with subclinical necrotic enteritis [34,35,36]. The dietary supplementation of live *Yeast* has the potential to alleviate lipopolysaccharide-induced inflammation in broilers [37]. In the present study, compound probiotics were also demonstrated to improve immune system function by increasing the concentrations of IgG, IgA, and IgM in the serum of broilers. This likely explains the increase in TP content in the serum of broilers that were fed the compound probiotics. *L. plantarum* 16 and *Paenibacillus polymyxa* 10 have been shown to increase intestinal barrier function, anti-oxidative capacity, and immunity, and decrease cell apoptosis [38]. In another study, *Lactobacillus* spp. reduced oxidative stress caused by deoxynivalenol on the intestine and liver of broilers [39]. In this study, the broilers that were fed compound probiotics also showed enhanced antioxidant capacity, evidenced by the increased activities of T-AOC, T-SOD, and GSH-Px and the decreased MDA content in their serum. The relative weight of the bursa and spleen in broilers was increased by the compound probiotics, which is consistent with a previous report showing that *Yeast* culture supplementation increased the relative organ weight of the bursa of Fabricius [18]. Taking all the evidence into consideration, we concluded that the compound probiotics could enhance the immune and antioxidant capacities of broilers, similar to other probiotics, while antibiotics lack such potency.

The intestinal health of broilers was improved by the dietary addition of *L. acidophilus*, *L. plantarum* 15-1, *L. plantarum* 16, or *L. plantarum* P8 via an increase in the concentrations of SCFAs (short chain fatty acids) and intestinal barrier function, a decrease in cell apoptosis, etc. [14,15,27,29,38]. *Yeast* and its derivatives can partially recover the intestinal health status of broilers with aflatoxin B1, necrotic enteritis, or coccidiosis [19,24]. *L. johnsonii* 3-1 and *L. crispatus* 7-4 were shown to promote ileum development in broilers [16]. *Lactobacillus* spp. reduced adverse morphological changes of the intestine induced by deoxynivalenol in broilers [39]. In the current study, the dietary supplementation of compound probiotics enhanced the intestinal absorptive capacity by increasing VH and decreasing CD. In addition, the compound probiotics decreased the amount of *E. coli* and increased the amount of *Lactobacillus* in the broilers’ cecum. This agrees with the results of previous studies reporting that L. acidophilus D2/CSL increased L. acidophilus counts, thus improving the population of beneficial microbes in the cecum of broilers [40], and that *Yeast* culture increased the prevalence of *Lactobacillus* in feces while decreasing that of *E. coli* [18]. In addition, *L. johnsonii* was also found to reduce *S. sofia* and *C. perfringens* in the gut and improve the colonization resistance of birds to *S. sofia* [41]. *Yeast* and its additives have also been shown to decrease the abundance of *Salmonella* in the cecum of broilers [33,42]. Therefore, we deduced that the compound probiotics enhanced the growth performance and health of broilers likely by improving their intestinal morphology and increasing the ratio of beneficial bacteria to pathogenic bacteria.

Not all diets formulated with probiotics benefit the intestinal microbiota structure in broilers [43]. In one study, the dietary supplementation of *Lactobacillus* and inulin showed positive effects on the gut microbiota of broilers [28]. The recombinant *Lactobacillus* bacteriocin plantaricin K was shown to adjust the distribution of the intestinal microbiome [44]. The dietary supplementation of *L. plantarum* P8 enhanced the growth performance and intestinal health of broilers infected with Eimeria by regulating their gut microbiota [15]. The addition of *L. acidophilus* contributed to the restoration of the microbial structure altered by C. perfringens infection [45]. The supplementation of a *Yeast*-based probiotic, including *Bacillus amyloliquefaciens*, *B. subtilis*, and *B. licheniformis*, enhanced the energy metabolism of cecal microbiota in young broilers [21]. *Yeast* cultures can obviously improve the growth performance of broilers by re-balancing their cecal microbiota [30]. In the present study, the cecal microbial diversity of broilers was also significantly changed by dietary compound probiotic supplementation; these changes were distinct from those of broilers fed antibiotics. In addition, the specific changes in the bacterial community in the cecum of broilers was also revealed using high-throughput sequencing including Bacteroidetes, Cyanobacteria, and Euryarchaeota at the phylum level, and *Alistipes*, *Burkholderia*, *Campylobacter*, *Coprobacillus*, *Eggerthella*, *Helicobacter*, *Lactococcus*, *Methanocorpusculum*, *Prevotella*, and *Sediminibacterium* at the genus level.

The intestinal microbiota exerts an influence on the intestinal and bodily health of the host mainly through the manipulation of microbial metabolites and direct interactions with intestinal epithelial cells. *Alistipes* is a newly discovered genus of bacteria that is closely related to dysbiosis and disease [46]. The genus *Burkholderia* usually thrives in harsh environments, and some of its members are remarkably opportunistic pathogens [47]. *Campylobacter* continues to be one of the most common bacteria causing gastroenteritis and diarrheal illness [48]. *Coprobacillus* is a potentially noxious bacteria associated with inflammation [49]. *Eggerthella* sp. DII-9 was identified and demonstrated to be a novel detoxification bacterium that degrades trichothecene mycotoxins [50]. *Enterohepatic Helicobacter* species have been demonstrated to be associated with human digestive diseases [51]. *Lactococcus* species are fully proven as microbiota that reflect common dietary contaminants [52]. *Methanocorpusculum labreanum* is common among hindgut fermenters of horses [53]. The increased abundance of *Prevotella* strains, a dominant bacterial genus in human gut microbial communities, is linked with dietary fiber-induced improvement in glucose metabolism [54]. The functions of Sediminibacterium strains in the intestine are still not clear. In this study, the compound probiotics significantly decreased the abundance of *Alistipes*, *Burkholderia*, *Coprobacillus*, *Helicobacter*, *Lactococcus*, and *Sediminibacterium* and increased that of *Campylobacter*, *Eggerthella*, and *Prevotella* in the cecal microbiota of broilers, relative to the NC or PC. Therefore, we deduced that the improvement in growth performance and health due to the effects of the compound probiotic could be attributed to the distinct positive alteration of the cecal microbiota of the broiler birds, including an increase in healthy bacteria genera, e.g., *Eggerthella*, and a decrease in pernicious bacteria, including *Alistipes*, *Burkholderia*, *Coprobacillus*, *Helicobacter*, and *Lactococcus*. The increase in *Campylobacter* may be an issue of concern once LY is used in broilers’ diets.

## 5. Conclusions

Our compound probiotics containing *Lactobacillus* and *Yeast* are a competent alternative to synthetic antibiotics as a growth and carcass enhancer for broilers. They can improve the immune and antioxidant capacities of broilers, similar other probiotics, while antibiotics lack such merit. The compound probiotics improved the growth performance and health of broilers, likely by improving their intestinal morphology and increasing the ratio of beneficial to pathogenic bacteria. The cecal microbial diversity of broilers was significantly changed by dietary compound probiotic supplementation, and these changes were distinct from those in broilers fed antibiotics. Specifically, the compound probiotics significantly decreased the abundance of *Alistipes*, *Helicobacter*, and *Lactococcus*, and increased the prevalence of *Campylobacter*, compared with the negative control group. Relative to the antimicrobial group, the compound probiotics significantly decreased the abundance of *Burkholderia*, *Coprobacillus*, and *Sediminibacterium*, and increased that of *Eggerthella* and *Prevotella*. These results indicate the distinct mechanism through which probiotics and antibiotics enhance the growth performance and health of broilers from the perspective of intestinal microbial structure, while also enriching the basic knowledge surrounding the intestinal microbial health of broilers.

## Figures and Tables

**Figure 1 biology-11-00633-f001:**
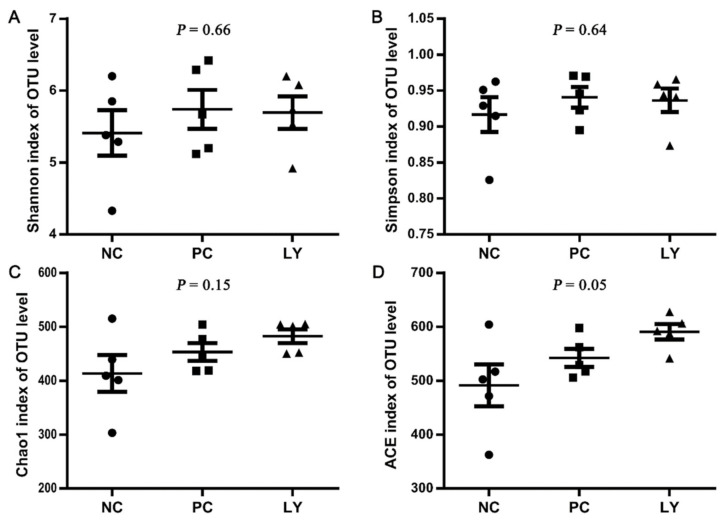
Cecal microbial α-diversity of broilers indicated by Shannon (**A**), Simpson (**B**), Chao1 (**C**), ACE (**D**) indexes of operational taxonomic unit (OTU) level. NC—negative control, basal diets; PC—positive control, basal diets supplemented with antimicrobials; LY—basal diets supplemented with compound probiotics. Each plot, including the dots, squares, and triangles, in the figure represents one sample.

**Figure 2 biology-11-00633-f002:**
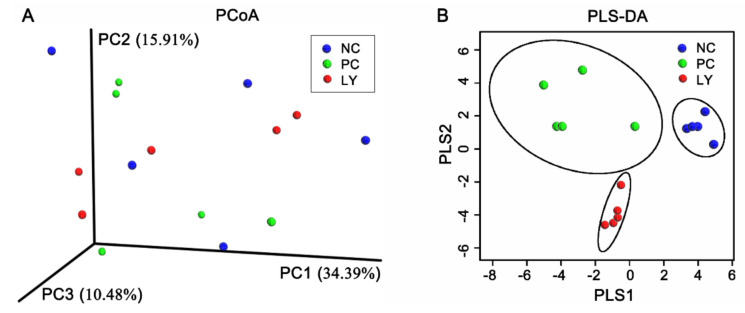
Microbial β-diversity in the cecum of broilers. (**A**) Principal coordinate analysis (PCoA) of weighted distance calculated from operational taxonomic unit (OTU) abundance matrix. (**B**) Partial least squares discriminant analysis (PLS-DA). NC—negative control, basal diets; PC—positive control, basal diets supplemented with antimicrobials; LY—basal diets supplemented with compound probiotics.

**Figure 3 biology-11-00633-f003:**
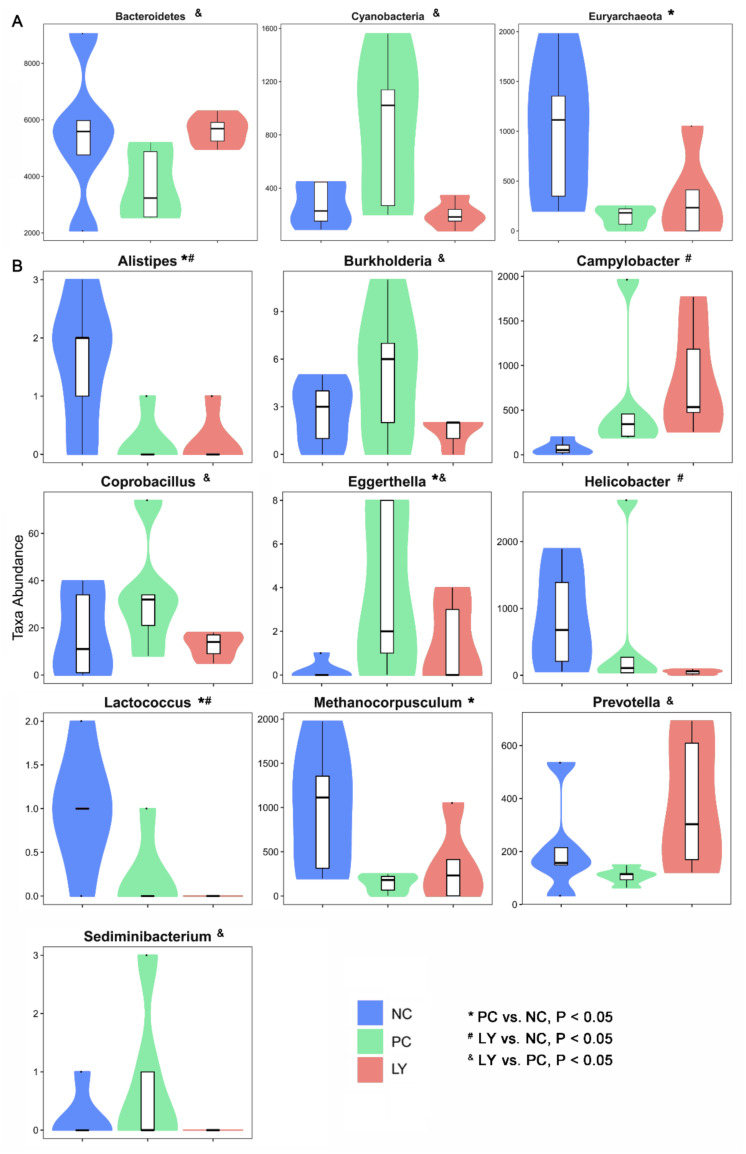
Different species at the phylum (**A**) and genus (**B**) level in the cecal microbiota of broilers. NC—negative control, basal diets; PC—positive control, basal diets supplemented with antimicrobials; LY—basal diets supplemented with compound probiotics. * significant difference (*p* < 0.05) between PC and NC groups. ^#^ significant difference (*p* < 0.05) between LY and NC groups. ^&^ significant difference (*p* < 0.05) between LY and PC groups.

**Table 1 biology-11-00633-t001:** Dietary composition and nutrient level of the basal diets.

Items	Basal Diets	
	Day 1 to 21	Day 22 to 42
Ingredients, %		
Corn	56.70	59.95
Soybean meal	31.80	27.55
Rapeseed meal	2.50	2.50
Cottonseed meal	2.00	2.50
Soybean oil	2.88	3.31
CaHPO_4_	1.90	1.85
Limestone	1.28	1.43
Salt	0.30	0.30
Choline chloride (50%)	0.10	0.10
DL-methionine (99%)	0.18	0.12
L-lysine HCl (78%)	0.14	0.17
Vitamin/mineral premix ^1^	0.22	0.22
Total	100.00	100.00
Nutrient level ^2^		
AME, MJ/kg	12.15	12.55
Crude protein, %	21.00 (20.89)	19.50 (19.03)
Calcium, %	1.00 (0.95)	0.90 (0.86)
Available phosphorus, %	0.48	0.40
Lysine, %	1.15 (1.13)	1.00 (0.98)
Methionine, %	0.50 (0.51)	0.40 (0.39)
Methionine + cystine, %	0.90 (0.89)	0.78 (0.75)
Threonine, %	0.88 (0.86)	0.75 (0.76)
Tryptophan, %	0.24 (0.23)	0.22 (0.22)

^1^ Premix supplied per kg of diet: biotin—0.0325 mg; Ca-pantothenate—12 mg; folic acid—1.25 mg; niacin—50 mg; vitamin A—12,500 IU; vitamin B_1_—2 mg; vitamin B_2_—6 mg; vitamin B_12_—0.025 mg; vitamin D_3_—2500 IU; vitamin E—18.75 IU; vitamin K_3_—2.65 mg; Cu—8 mg; Fe—80 mg; I—0.35 mg; Mn—100 mg; Se—0.15 mg; Zn—75 mg. ^2^ Nutrient levels listed are calculated values. Numbers in parenthesis are analyzed values.

**Table 2 biology-11-00633-t002:** Effect of dietary treatments on the growth performance of broilers.

Items	NC	PC	LY	SEM	*p*-Value
Body weight, g					
Day 0	47.58	47.47	47.61	0.29	0.983
Day 21	794.54 ^b^	816.59 ^ab^	841.29 ^a^	6.48	0.005
Day 42	2364.86 ^c^	2453.05 ^b^	2614.57 ^a^	26.96	<0.001
Day 1 to 21					
ADFI, g	48.21	48.34	47.71	0.29	0.763
ADG, g	35.76 ^b^	36.72 ^b^	38.12 ^a^	0.32	0.003
FCR, g/g	1.35 ^a^	1.32 ^a^	1.25 ^b^	0.06	0.003
Day 22 to 42					
ADFI, g	139.78 ^b^	144.35 ^ab^	148.20 ^a^	1.38	0.033
ADG, g	73.54 ^c^	76.81 ^b^	83.59 ^a^	1.14	<0.001
FCR, g/g	1.90 ^a^	1.88 ^a^	1.77 ^b^	0.02	0.002
Day 1 to 42					
ADFI, g	92.96	94.96	96.44	0.68	0.098
ADG, g	54.62 ^c^	56.65 ^b^	60.38 ^a^	0.64	<0.001
FCR, g/g	1.70 ^a^	1.68 ^a^	1.60 ^b^	0.01	<0.001

ADFI—average feed intake; ADG—average daily gain; FCR—feed conversion ratio, feed/gain, g/g; NC—negative control, basal diets; PC—positive control, basal diets supplemented with antimicrobials; LY—basal diets supplemented with compound probiotics. ^a, b, c^ Means with no common superscript in the same line differ significantly (*n* = 6, *p* < 0.05).

**Table 3 biology-11-00633-t003:** Effect of dietary treatments on the serum biochemical indexes of broilers.

Items	NC	PC	LY	SEM	*p*-Value
Day 21					
ALB, g/L	11.86	12.58	12.75	0.18	0.109
ALT, U/L	3.22	3.00	2.90	0.15	0.692
AST, U/L	255.44	241.64	229.50	11.91	0.702
CRE, umol/L	40.84	42.87	41.30	1.32	0.811
TP, g/L	25.23	26.36	26.70	0.51	0.498
UA, umol/L	491.67	494.45	448.30	9.64	0.085
Day 42					
ALB, g/L	14.27	14.58	14.56	0.22	0.825
ALT, U/L	4.77	3.90	4.30	0.23	0.304
AST, U/L	374.44	359.44	357.90	5.58	0.447
CRE, umol/L	57.74	61.70	60.26	1.65	0.637
TP, g/L	30.59 ^b^	30.37 ^b^	32.63 ^a^	0.42	0.044
UA, umol/L	457.33	427.27	348.70	21.96	0.120

ALB—albumin; ALT—alanine aminotransferase; AST—aspartate aminotransferase; CRE—creatinine; TP—total protein; UA—uric acid; NC—negative control, basal diets; PC—positive control, basal diets supplemented with antimicrobials; LY—basal diets supplemented with compound probiotics; ^a, b^ Means with no common superscript in the same line differ significantly (*n* = 6, *p* < 0.05).

**Table 4 biology-11-00633-t004:** Effect of dietary treatments on the serum antioxidant indexes of broilers.

Items	NC	PC	LY	SEM	*p*-Value
Day 21					
GSH-P_x_, 10^3^ U/ml	1.58 ^b^	1.54 ^b^	1.78 ^a^	0.03	<0.001
MDA, nmol/mL	5.81 ^a^	4.30 ^b^	3.38 ^b^	0.32	0.003
T-AOC, U/mL	7.22 ^b^	7.44 ^b^	11.59 ^a^	0.46	<0.001
T-SOD, U/mL	81.74 ^c^	97.07 ^b^	107.85 ^a^	2.93	<0.001
Day 42					
GSH-P_x_, 10^3^ U/mL	1.35 ^b^	1.37 ^b^	1.95 ^a^	0.07	<0.001
MDA, nmol/mL	4.27 ^a^	3.32 ^ab^	2.63 ^b^	0.23	0.010
T-AOC, U/mL	9.12	10.11	10.72	0.44	0.344
T-SOD, U/mL	108.33 ^b^	108.78 ^b^	124.01 ^a^	2.17	0.001

GSH-Px—glutathione peroxidase; MDA—malondialdehyde; T-AOC—total antioxidant capacity; T-SOD—total superoxide dismutase; NC—negative control, basal diets; PC—positive control, basal diets supplemented with antimicrobials; LY—basal diets supplemented with compound probiotics. ^a, b, c^ Means with no common superscript in the same line differ significantly (*n* = 6, *p* < 0.05).

**Table 5 biology-11-00633-t005:** Effect of dietary treatments on the serum immune index of broilers.

Items, µg/mL	NC	PC	LY	SEM	*p*-Value
Day 21					
IgA	0.88 ^b^	0.9 ^b^	1.24 ^a^	0.10	<0.001
IgG	7.96 ^b^	7.78 ^b^	9.91 ^a^	0.24	<0.001
IgM	1.32 ^b^	1.36 ^b^	1.69 ^a^	0.08	<0.001
Day 42					
IgA	1.11 ^b^	1.03 ^b^	1.36 ^a^	0.13	0.001
IgG	6.87 ^b^	6.64 ^b^	10.05 ^a^	0.52	<0.001
IgM	1.03 ^b^	1.08 ^b^	1.46 ^a^	0.14	<0.001

Ig—immune globulin; NC—negative control, basal diets; PC—positive control, basal diets supplemented with antimicrobials; LY—basal diets supplemented with compound probiotics. ^a, b^ Means with no common superscript in the same line differ significantly (*n* = 6, *p* < 0.05).

**Table 6 biology-11-00633-t006:** Effect of dietary treatments on the immune organ index of broilers.

Items, %	NC	PC	LY	SEM	*p*-Value
Day 21					
Bursa	2.00	1.87	2.17	0.23	0.064
Spleen	1.01	0.92	1.11	0.18	0.108
Thymus	2.11	2.13	2.17	0.37	0.942
Day 42					
Bursa	0.49 ^b^	0.50 ^b^	0.54 ^a^	0.01	<0.001
Spleen	1.04 ^b^	0.98 ^b^	1.16 ^a^	0.05	<0.001
Thymus	1.40	1.40	1.44	0.20	0.209

NC—negative control, basal diets; PC—positive control, basal diets supplemented with antimicrobials; LY—basal diets supplemented with compound probiotics. ^a, b^ Means with no common superscript in the same line differ significantly (*n* = 6, *p* < 0.05).

**Table 7 biology-11-00633-t007:** Effect of dietary treatments on the carcass traits of broilers.

Items, %	NC	PC	LY	SEM	*p*-Value
Eviscerated yield	72.98	73.37	74.87	0.36	0.074
Breast muscle	25.75 ^b^	26.81 ^a^	26.82 ^a^	0.17	0.006
Leg muscle	19.61 ^b^	20.16 ^b^	20.77 ^a^	0.15	0.003
Abdominal fat	1.44 ^a^	1.56 ^ab^	1.39 ^b^	0.03	0.050

NC—negative control, basal diets; PC—positive control, basal diets supplemented with antimicrobials; LY—basal diets supplemented with compound probiotics. ^a, b^ Means with no common superscript in the same line differ significantly (*n* = 6, *p* < 0.05).

**Table 8 biology-11-00633-t008:** Effect of dietary treatments on the jejunum morphology of broilers.

Items	NC	PC	LY	SEM	*p*-Value
Day 21					
Crypt depth (μm)	310.32 ^a^	311.27 ^a^	268.73 ^b^	6.22	0.001
Villus height (μm)	1025.60 ^b^	1084.30 ^b^	1213.57 ^a^	28.59	0.012
VCR	3.31 ^b^	3.49 ^b^	4.51 ^a^	0.14	<0.001
Day 42					
Crypt depth (μm)	282.38	270.16	261.30	4.25	0.123
Villus height (μm)	1174.50 ^b^	1197.20 ^b^	1411.20 ^a^	33.25	0.001
VCR	4.17 ^b^	4.44 ^b^	5.40 ^a^	0.14	<0.001

VCR—the ratio of villus height and crypt depth; NC—negative control, basal diets; PC—positive control, basal diets supplemented with antimicrobials; LY—basal diets supplemented with compound probiotics. ^a, b^ Means with no common superscript in the same line differ significantly (*n* = 6, *p* < 0.05).

**Table 9 biology-11-00633-t009:** Effect of dietary treatments on the cecum microbiota of broilers.

Items, CFU/g	NC	PC	LY	SEM	*p*-Value
Day 21					
*Escherichia coli*	7.55	7.32	7.35	0.52	0.784
*Lactobacillus*	7.40	7.32	7.68	0.36	0.431
*Escherichia coli*/*Lactobacillus*	1.02	1.00	0.96	0.06	0.246
Day 42					
*Escherichia coli*	7.86 ^a^	6.90 ^b^	6.18 ^c^	0.47	<0.001
*Lactobacillus*	7.55 ^b^	7.60 ^b^	8.04 ^a^	0.17	0.002
*Escherichia coli*/*Lactobacillus*	1.04 ^a^	0.91 ^b^	0.77 ^c^	0.07	<0.001

NC—negative control, basal diets; PC—positive control, basal diets supplemented with antimicrobials; LY—basal diets supplemented with compound probiotics. ^a, b, c^ Means with no common superscript in the same line differ significantly (*n* = 6, *p* < 0.05).

## Data Availability

Data from the current study are presented in the article/Appendix A. The raw data are available by contacting the corresponding authors.

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
