# Peer review of "Dietary Supplementation of a New Probiotic Compound Improves the Growth Performance and Health of Broilers by Altering the Composition of Cecal Microflora"

_biology, 2022, doi:10.3390/biology11050633_

Round 1
Reviewer 1 Report
The research field is important for poultry and human health.
The title is misleading since balancing of cecal microflora as the reason behind the improved performance was not tested in the study (it is a hypothesis based on the findings)
Methods
The probiotic compound is not fully described in the methods-what is it? Where was the source? what are 1X and 2 X?
Is Virginiamycin the prototype antimicrobial used in broiler feeds or why was it selected
Was it given all through 42 days of the experiment
How were the tissues collected, processed for histology and other parameters shown in the results
Results
The information in the text does not agree with table numbering. The number of animals sacrificed at the end of the experiment is inconsistent with n values mentioned in the table n=6??
The probiotics tested increased the number of Campylobacter-is this a desirable outcome?
Author Response
Dear reviewer,
Special thanks to you for your scholarly comments on our manuscript. We have studied the comments carefully and have made the necessary corrections. We used the track changes in MS Word during the revision. The one-by one responses to your comments are as follows. If you have any question, please don't hesitate to contact me. Thank you for your kindly supporting.
Q1: The title is misleading since balancing of cecal microflora as the reason behind the improved performance was not tested in the study (it is a hypothesis based on the findings)
Answer: Just as you said, it was not confirmed. So we corrected it into “may through” in the title.
Methods
Q2: The probiotic compound is not fully described in the methods-what is it? Where was the source? what are 1X and 2 X?
Answer: The probiotics used in the present study were explored by our research group. In order to keep the copyright of the products, we used x1 and x2 to represent their strains. We supplemented related explanation in the manuscript.
Q3: Is Virginiamycin the prototype antimicrobial used in broiler feeds or why was it selected
Answer: Yes. virginiamycin is the antibiotic growth promoter commonly used in broiler feeds. In the current study, virginiamycin was selected to evaluate the effects of compound probiotics on the substitution of antibiotic growth promoters.
Q4: Was it given all through 42 days of the experiment
Answer: Yes. The feeding trial lasted for 42 days. We supplemented the information in the section of “Animals and Experimental Diets”.
Q5: How were the tissues collected, processed for histology and other parameters shown in the results
Answer: We are sorry for the missing content. We supplemented one section “Jejunum Histomorphology” and related content in the section “Data and Sample Collection” as follows.
About 2 cm of jejunum (medial portion posterior to the bile ducts and anterior to Meckel’s diverticulum) were cut off gently and fixed in 10% formalin for histomorphology.
2.3. Jejunum Histomorphology
Fixed jejunum samples were embedded in paraffin, making histological sections and staining with hematoxylin and eosin. Three sections cutting vertically from villus enterocytes to the muscularis mucosa were selected for one sample. The vertical distance from the villus tip to villus–crypt junction level was taken as intestinal villus height (VH), and the vertical distance from the villus-crypt junction to the lower limit of the crypt as crypt depth (CD). Ten loci per section was measured for VH and CD. The ratio of VH and CD was calculated as V/C. The microscope coupled with a Microcomp integrated digital imaging analysis system (Nikon Eclipse 80i, Nikon Co., Tokyo, Japan) was used for the measurement.
Results
Q6: The information in the text does not agree with table numbering. The number of animals sacrificed at the end of the experiment is inconsistent with n values mentioned in the table n=6??
Answer: In fact, one bird per replicate was slaughtered at day 21 and 42, respectively. We corrected the description as follows.
One bird that closed to the average BW of the replicate were selected from each replicate at day 21 and 42, respectively, for blood sampling and carcass measurement after a 12 hours fast.
Q7: The probiotics tested increased the number of Campylobacter-is this a desirable outcome?
Answer: It is not a desirable outcome. In the discussion section, we said “Campylobacter continues to be one of the most common bacterial causes of gastroenteritis and diarrheal illness (48).” But at end of the section, our description was not suitable as “Therefore, we deduced that the improvement of compound probiotics on growth performance and body health probably eventually ascribed the special changes of cecal microbiota of broilers, including the increase of healthy bacteria genera and the decrease of pernicious ones.”. Now, we detailed it as follows.
Therefore, we deduced that the improvement of compound probiotics on growth performance and body health probably eventually ascribed the special changes of cecal microbiota of broilers, including the increase of healthy bacteria, Eggerthella and Prevotella and the decrease of pernicious ones, Alistipes, Burkholderia, Coprobacillus, Helicobacter, and Lactococcus.
Reviewer 2 Report
The manuscript deals with the results of an in vivo study related to the effects of certain probiotics as feed supplements on the growth and health status of broiler chickens. The trial is generally well designed and conducted, and the performed analyses provide a comprehensive overview of the growth performance, carcass yield, immune status and gut health of the chickens reared on a diet supplemented with a combination of Lactobacilli and yeast. However, I have the following concerns to be addressed before acceptance.
Major concerns:
- It should be clearly specified, which species and strain was applied in case of both Lactobacillus and yeast. The exact specification of the applied probiotics is essential for publishing the data obtained. It should be stressed by the authors that the published data are specifically related to a certain Lactobacillus strain and yeast species, and not to probiotics in general. Please also modify the title to be more specific for the applied probiotics.
- The study is complex and well designed, the results are clearly presented, but its novelty may be questionable, so it should be justified by the authors. Several papers have been already published concerning the effects of different probiotics (among others, Lactobacilli and Saccharomyces spp. as well) on growth performance and gut health of broilers. However, the obtained results can be novel for the applied Lactobacillus strain and yeast species, and the high number of parameters investigated also increases the merit of the study. Please clearly indicate what is already known and what is new concerning the applied probiotics.
- Why was Lactobacillus and yeast supplementation applied only in combination, and not separately? It would have been better to also include separate groups with the two probiotics. Please comment on this in the manuscript as well.
- Statistics and data analysis: the obtained data are correctly evaluated, but it should be stated how the two samples per replicate were considered. In the statistics section, n=6/group is indicated referring to the 6 replicates, which is correct, but how was the value of each replicate gained? Is it the mean of the two samples within a replicate? Concerning data presentation, I would prefer to present individual SEMs for each group, but pooled SEM is also acceptable.
- Conclusion, lines 443-446., gut microbiota composition: I suggest to summarize here only such results, where a significant difference can be observed between LY (probiotics) and NC (absolute control) groups, and not between LY and PC (virginiamycin group). It is misleading that significant effects of the probiotic supplementation are listed here without indicating whether it was obtained compared to NC or PC. For instance, Burkholderia and Coprobacillus were influenced by LY only compared to the PC group, but they are listed together with LY vs. NC differences.
- Line 139: not all viable bacteria, just E. coli and Lactobacilli were determined, please modify the subheading accordingly.
Minor concerns:
- 1. and 3.: please indicate the type of the plots in the figure captions.
- Line 123-124.: please indicate which ingesta sample was used for which analysis. I guess that the shock-frozen one was used for qPCR and sequencing, and the other one stored on ice for culturing E. coli and Lactobacilli.
- Please indicate which software was used for analyzing the qPCR and sequencing data.
Author Response
Dear reviewer,
Special thanks to you for your scholarly comments on our manuscript. We have studied the comments carefully and have made the necessary corrections. We used the track changes in MS Word during the revision. The one-by one responses to your comments are as follows. If you have any question, please don't hesitate to contact me. Thank you for your kindly supporting.
Major concerns:
Q1: It should be clearly specified, which species and strain was applied in case of both Lactobacillus and yeast. The exact specification of the applied probiotics is essential for publishing the data obtained. It should be stressed by the authors that the published data are specifically related to a certain Lactobacillus strain and yeast species, and not to probiotics in general. Please also modify the title to be more specific for the applied probiotics.
Answer: The probiotics used in the present study were explored by our research group. In order to keep the copyright of the products, we used x1 and x2 to represent their strains. We supplemented related explanation in the manuscript. According to your suggestion, we modify the title as “Dietary supplementation of exclusive probiotics improve growth performance and body health of broilers may through balancing the composition of cecal microflora”.
Q2: The study is complex and well designed, the results are clearly presented, but its novelty may be questionable, so it should be justified by the authors. Several papers have been already published concerning the effects of different probiotics (among others, Lactobacilli and Saccharomyces spp. as well) on growth performance and gut health of broilers. However, the obtained results can be novel for the applied Lactobacillus strain and yeast species, and the high number of parameters investigated also increases the merit of the study. Please clearly indicate what is already known and what is new concerning the applied probiotics.
Answer: Thanks for your constructive concern. As you said that the effects of different probiotics on growth performance and gut health of broilers have been previously studied. In the introduction section, based on previous literatures, we pointed out that both of Lactobacillus and Yeast are suitable alternatives for antibiotic growth promoter in diets of broilers. As for the novelty of this study, on the one hand, multiple strains of probiotics could induce a "synergistic effect" that enhances the survival of probiotic bacteria and maintains the balance and stability of the host gas-trointestinal microbiota. The combination of Lactobacillus and Yeast is expected to achieve a better therapeutic result as adjuvant or alternative therapies for enhancing the antibacterial effects in broiler production. On the other hand, the strains of Lactobacillus and Yeast were newly and exclusively explored for broilers by our research group, and could remain viable in feed and through the adverse conditions of the upper gastrointestinal tract. Therefore, the current study is meaningful for exploring suitable alternatives for antibiotic growth promoter in diets of broilers.
Q3: Why was Lactobacillus and yeast supplementation applied only in combination, and not separately? It would have been better to also include separate groups with the two probiotics. Please comment on this in the manuscript as well.
Answer: Thank for your question and suggestion. Multiple strains of probiotics could enhances the survival of probiotic bacteria and maintains the balance of the intestinal microbiota. The stains of probiotics, Lactobacillus and yeast, were designed as a compound preparation from the beginning and explored as a new product to substitute the antibiotic. According to your suggestion, we supplemented related comments in the discussion section as follows.
Multiple strains of probiotics could enhances the survival of probiotic bacteria and maintains the balance of the intestinal microbiota (8). Both of Lactobacillus x1 and Yeast x2 used in the current study were newly selected as two potential probiotic strains that could remain vitality in feed and gastrointestinal tract, and designed as a compound preparation from the beginning to substitute antibiotics in broilers’ diets.
Q4: Statistics and data analysis: the obtained data are correctly evaluated, but it should be stated how the two samples per replicate were considered. In the statistics section, n=6/group is indicated referring to the 6 replicates, which is correct, but how was the value of each replicate gained? Is it the mean of the two samples within a replicate? Concerning data presentation, I would prefer to present individual SEMs for each group, but pooled SEM is also acceptable.
Answer: In fact, one bird per replicate was slaughtered at day 21 and 42, respectively. We corrected the description as follows.
One bird that closed to the average BW of the replicate were selected from each replicate at day 21 and 42, respectively, for blood sampling and carcass measurement after a 12 hours fast.
Q5: Conclusion, lines 443-446., gut microbiota composition: I suggest to summarize here only such results, where a significant difference can be observed between LY (probiotics) and NC (absolute control) groups, and not between LY and PC (virginiamycin group). It is misleading that significant effects of the probiotic supplementation are listed here without indicating whether it was obtained compared to NC or PC. For instance, Burkholderia and Coprobacillus were influenced by LY only compared to the PC group, but they are listed together with LY vs. NC differences.
Answer: We are sorry for the misleading description. Both of the significant differences between LY and NC groups, and between LY and PC, are meaningful for the evaluation of compound probiotics. In order to make the conclusion more clear, we detailed the description in the discussion section as follows indicating whether it was obtained compared to NC or PC.
In specific, the compound probiotics significantly decreased the abundance of Alistipes, Helicobacter, and Lactococcus, and increased Campylobacter, compared with the negative group. Relative to the antimicrobial group, the compound probiotics significantly decreased Burkholderia, Coprobacillus, and Sediminibacterium, increased Eggerthella and Prevotella.
Q6: Line 139: not all viable bacteria, just E. coli and Lactobacilli were determined, please modify the subheading accordingly.
Answer: We corrected the sub-title into “Counts of Viable E. coli and Lactobacilli” and also modified the under description.
Minor concerns:
Q7: 1. and 3.: please indicate the type of the plots in the figure captions.
Answer: We supplemented related information in the figure captions, according to your suggestion.
Q8: Line 123-124.: please indicate which ingesta sample was used for which analysis. I guess that the shock-frozen one was used for qPCR and sequencing, and the other one stored on ice for culturing E. coli and Lactobacilli.
Answer: Your guess is right. We detailed the description in the manuscript.
Q9: Please indicate which software was used for analyzing the qPCR and sequencing data.
Answer: We supplemented related information according to your suggestion.
Reviewer 3 Report
Dear Authors,
the manuscript is an interesting scientific study. Below are my comments:
- abstract is, according to my calculations, too long (not in accordance with the requirements of the journal)
- citation, not in accordance with the requirements. It should be, for example, [8] and not (8).
- introduction - correctly described, introduces the reader to the discussed research issues.
- the birds were kept for 42 days with 23 hours of light? Why has the light level not been lowered? This is unethical!
- please describe in more detail the housing conditions of the birds.
- intestinal content was collected. How was it carried out? Before slaughter, the birds were fasted for 12 hours, was there much content after such a fast?
- section 2.3. - I am asking for a more detailed description, and not just referring to other sources of literature.
- did all the data collected in the research meet the criteria of parametric analysis? The normal distribution and homogeneity of the samples were checked? How?
- Section 3 description - correct.
- e.g. BWG = 2364.86 - 47.58 = 2317.28.
2317.28 / 42 days = 55.17 (ADG)? How was ADG ADFI calculated? Please provide the formulas in the methodology. - the NC showed a 72.98% yield and the LY 74.87% yield. There were no statistically significant differences here? Definitely?
- the discussion is correct. I can only suggest that the mechanisms could be described in more detail. How the substance influenced the chickens to weigh more, etc.
- In the application, please add an application for the internship
The entire work requires editorial correction, in accordance with the instructions for authors.
I recommend improving the manuscript.
Yours sincerely
Author Response
Dear reviewer,
Special thanks to you for your scholarly comments on our manuscript. We have studied the comments carefully and have made the necessary corrections. We used the track changes in MS Word during the revision. The one-by one responses to your comments are as follows. If you have any question, please don't hesitate to contact me. Thank you for your kindly supporting.
Q1: abstract is, according to my calculations, too long (not in accordance with the requirements of the journal)
Answer: According to your suggestion, we simplified the abstract section into 200 words in accordance with the requirements of the journal.
Q2: citation, not in accordance with the requirements. It should be, for example, [8] and not (8).
Answer: We corrected it, and checked all citations throughout the manuscript.
Q3: introduction - correctly described, introduces the reader to the discussed research issues.
Answer: Thank you for your approval.
Q4: the birds were kept for 42 days with 23 hours of light? Why has the light level not been lowered? This is unethical!
Answer: We are also concerned about animal welfare issues. Bird management in the current study was consistent with recommendations of the AA Broiler Management Guide (Aviagen, 2009). The 23L:1D light regime is widely used in commercial broiler industry. The study was approved by the Animal Care and Use Committee of the Institute of Feed Research of the Chinese Academy of Agricultural Sciences (Beijing, China). We will further strengthen the protection of animal welfare in future research.
Q5: please describe in more detail the housing conditions of the birds.
Answer: We supplemented related information according to your suggestion.
Q6: intestinal content was collected. How was it carried out? Before slaughter, the birds were fasted for 12 hours, was there much content after such a fast?
Answer: The birds was not slaughtered immediately after blood sampling. Before slaughter, the birds were offered the experimental diets for 6 h. We detailed the description in the manuscript.
Q7: section 2.3. - I am asking for a more detailed description, and not just referring to other sources of literature.
Answer: We detailed the description according to your suggestion.
Q8: did all the data collected in the research meet the criteria of parametric analysis? The normal distribution and homogeneity of the samples were checked? How?
Answer: All the data were checked for the normal distribution and homogeneity using Shapiro-Wilk test. We supplemented related information according to your suggestion.
Q9: Section 3 description - correct.
e.g. BWG = 2364.86 - 47.58 = 2317.28.
2317.28 / 42 days = 55.17 (ADG)? How was ADG ADFI calculated? Please provide the formulas in the methodology.
Answer: ADG=BWG/days. ADFI= FI/days. The reason for the above small difference is that we slaughtered a bird in each replicate at 21 days. We supplemented the formulas in the methods according to your suggestion as follows.
Average daily feed intake (ADFI, feed intake: days, g: day), average daily gain (ADG, BW gain: days, g: day) and feed conversion ratio (FCR, feed: gain, g: g) were calculated in replicates accordingly.
Q10: the NC showed a 72.98% yield and the LY 74.87% yield. There were no statistically significant differences here? Definitely?
Answer: We checked the result of data analysis. The P-value between NC and LY is 0.053, showing no statistically significant differences. Thank you for your concern. The Numerical differences are important implications for production. We will continue to pay attention on it in the pilot-scale test in the further.
Q11: the discussion is correct. I can only suggest that the mechanisms could be described in more detail. How the substance influenced the chickens to weigh more, etc.
Answer: We are carrying out another study on the underlying mechanism from the perspective of microbial metabolites, plasma metabolomics, Proliferation of muscle stem cells, etc. Please pay attention to our next report. Thank you.
Q12: In the application, please add an application for the internship
Answer: Thanks for your reminding. We will carry out a pilot-scale test in commercial farm before the large-scale application.
Q13: The entire work requires editorial correction, in accordance with the instructions for authors.
Answer: We formatted our manuscript following the Journal.
Round 2
Reviewer 1 Report
The title is vague-what are exclusive probiotics
The title needs rewriting-English editing
In the absence of a detailed explanation of the relevance of the changes in microflora observed (from literature) either increased or decreased in the host chicken (broiler), it is not appropriate to imply the observed changes reflect a balanced chicken cecum.
Increased Campy I would argue is not a balanced cecum, the only potential improved microflora difference between LY and NC in microflora is Eggerthella. The discussion needs other potential explanations of the observed differences.
Lines 323-326 do not agree with the results in figure 3. Fig 3 legend does not agree with other information in the box, the meaning of #,& and *
Line 468-470 still refers to Campy as healthy bacteria??
The undesirable outcome of increased Campy still needs to be addressed further in the discussion since it negates any benefits associated with a feed supplement from a public health point of view. How do the authors hope to address this? What is the explanation?
Also, what is the relevance of the improved microflora in the ceaca to the chicken digestive system either from past or new research. The inclusion of this information in the discussion in relation to the study findings may qualify the conclusion of the study.
This statement: "These findings demonstrated the feasibility of compound
probiotics to replace antibiotics in broilers’ diets" seems not to be supported by study findings unless qualified by how the authors wish to address the Campy increased concerns with the tested probiotics.
Author Response
Dear reviewer,
Special thanks to you for your scholarly comments on our manuscript. We have studied the comments carefully and improved our manuscript. We used the track changes in MS Word during the revision. The one-by one responses to your comments are as follows. If you have any question, please don't hesitate to contact me. Thank you for your kindly supporting.
Q1: The title is vague-what are exclusive probiotics
The title needs rewriting-English editing
Answer: Thanks for your concern. We corrected “exclusive probiotics” into “a new probiotic compound preparation” in the title.
Q2: In the absence of a detailed explanation of the relevance of the changes in microflora observed (from literature) either increased or decreased in the host chicken (broiler), it is not appropriate to imply the observed changes reflect a balanced chicken cecum.
Answer: We corrected it into “altering”.
Q3: Increased Campy I would argue is not a balanced cecum, the only potential improved microflora difference between LY and NC in microflora is Eggerthella. The discussion needs other potential explanations of the observed differences.
Answer: Thanks for your suggestion. We agree with you that Campylobacter is one of the most common bacteria resulting gastroenteritis and diarrheal illness and bad for intestinal health of broilers. We corrected and improved the description in the discussion section.
Q4: Lines 323-326 do not agree with the results in figure 3. Fig 3 legend does not agree with other information in the box, the meaning of #,& and *
Answer: I am sorry for the mistake. We corrected the figure legend.
Q5: Line 468-470 still refers to Campy as healthy bacteria??
Answer: I am sorry for the omission during the last round of revision. We corrected the description.
Q6: The undesirable outcome of increased Campy still needs to be addressed further in the discussion since it negates any benefits associated with a feed supplement from a public health point of view. How do the authors hope to address this? What is the explanation?
Answer: The compound probiotics could improve body immune, antioxidant capacities, growth performance and carcass traits of broilers. It is a competent alternative of synthetic antibiotics. But Campylobacter increased by LY is a bad bacteria from either broiler production or a public health point of view. Therefore, The increase of Campylobacter should be concerned once LY is used in broilers’ diets. We supplemented related explanation in the discussion section.
Q7: Also, what is the relevance of the improved microflora in the caeca to the chicken digestive system either from past or new research. The inclusion of this information in the discussion in relation to the study findings may qualify the conclusion of the study.
Answer: Thanks for your constructive suggestion. Based on previous reports, we deducted that microbiota enhance intestine and body health mainly through microbial metabolites and direct cell interactions. In the further study, we will analyze the metabolites of the key microbiota, function of the intestinal epithelial cells, and composition of plasma metabolite, then try to explore the underlying mechanism. As you suggested, we enriched the discussion to support the conclusion in the revised manuscript.
Q8: This statement: "These findings demonstrated the feasibility of compound
probiotics to replace antibiotics in broilers’ diets" seems not to be supported by study findings unless qualified by how the authors wish to address the Campy increased concerns with the tested probiotics.
Answer: Thanks for your concern. The increase of Campylobacter is indeed a problem that should be solved when using LY in broilers’ diets, but it does not stop us from saying LY is a good alternative to antibiotics. The explanation is shown as follows. The purpose of adding antibiotics to broiler feed is to promote growth performance, but it has many negative effects such as destroying intestinal function, increasing microbial antibiotic resistance and antibiotic residues in chicken. The probiotics in this study have been proven to have a good growth-promoting effect. Probiotics do not increase microbial resistance and drug residues in chicken, which is the biggest advantage over antibiotics. The increase of Campylobacter in the intestine is a negative side-effect of the LY, which will be considered, well studied and then removed may through inserting a plasmid with the resistance gene against Campylobacter into Lactobacillus or Yeast during the further studies.
Reviewer 2 Report
The authors have carefully addressed my concerns, substantially improved the manuscript and provided detailed answers to my questions. However, there is a major issue reamining, which has to be corrected before being accepted for publication. As I wrote in my previous report, the applied Lactobacillus and yeast species and strains have to be clearly specified. I can understand the copyright issues the authors have mentioned in the response letter, but in my opinion, presenting the exact species and strain of the applied probiotics is essential for publishing the results. Indicating the probiotic strains with just #1 and #2 has no meaning, it is absolutely not enough. Without exactly indicating the used species and strain, the readers do not know, for which specific Lactobacillus and yeast the results refer to, it remains completely unknown, which probiotic candidates were applied. Hence, I strongly suggest to include more data on the applied probiotics. In my opinion, the manuscript cannot be accepted for publication without providing the requested specific information of the probiotics.
Further, the modified version of the title is confusing, and it also needs some grammatical improvements, please modify it.
Author Response
Dear reviewer,
Thanks for your comments. The information of probiotics strains are supplemented according to your suggestion. We revised our manuscript for grammatical improvements with the help of an English native speaker. Thank you for your kindly supporting.
Reviewer 3 Report
Dear Authors,
thank you for the improvement of the manuscript.
I have no more questions.
Kind Regards.
Author Response
Thanks for your kind help and support.
Round 3
Reviewer 1 Report
The title is still not clear. Replace "may" with 'potentially' Have an English reader go through the manuscript to make the content clearer and correct grammar.
Author Response
Dear reviewer,
Thanks for your comments on our manuscript. We revised our manuscript to make its content clearer and correct grammar with the help of an English native speaker. If you have any question, please don't hesitate to contact me. Thank you for your kindly supporting.
Reviewer 2 Report
The authors have added the required specification of the probiotic strains used, so the revised version now contains sufficient information about the applied probiotics. The authors also modified the title as I have suggested. Hence, I have no more comments, I suggest to accept the manuscript for publication.